# A molecular photosensitizer achieves a $V_{oc}$ of 1.24 V enabling highly efficient and stable dye-sensitized solar cells with copper(II/I)-based electrolyte

Dan Zhang[1,2,5], Marko Stojanovic [1,5], Yameng Ren [1,5✉], Yiming Cao[1,4], Felix T. Eickemeyer [1,2✉], Etienne Socie [3], Nick Vlachopoulos[2], Jacques-E. Moser [3], Shaik M. Zakeeruddin[1], Anders Hagfeldt [2✉] & Michael Grätzel [1✉]

To develop photosensitizers with high open-circuit photovoltage ($V_{oc}$) is a crucial strategy to enhance the power conversion efficiency (PCE) of co-sensitized solar cells. Here, we show a judiciously tailored organic photosensitizer, coded MS5, featuring the bulky donor N-(2′,4′-bis (dodecyloxy)-[1,1′-biphenyl]-4-yl)-2′,4′-bis(dodecyloxy)-N-phenyl-[1,1′-biphenyl]-4-amine and the electron acceptor 4-(benzo[c][1,2,5]thiadiazol-4-yl)benzoic acid. Employing MS5 with a copper (II/I) electrolyte enables a dye-sensitized solar cell (DSC) to achieve a strikingly high $V_{oc}$ of 1.24 V, with the $V_{oc}$ deficit as low as 130 mV and an ideality factor of merely 1.08. The co-sensitization of MS5 with the wider spectral-response dye XY1b produces a highly efficient and stable DSC with the PCE of 13.5% under standard AM1.5 G, 100 mW cm$^{-2}$ solar radiation. Remarkably, the co-sensitized solar cell (active area of 2.8 cm$^2$) presents a record PCE of 34.5% under ambient light, rendering it very attractive as an ambient light harvesting energy source for low power electronics.

[1] Laboratory of Photonics and Interfaces, Institute of Chemical Sciences & Engineering, École Polytechnique Fédérale de Lausanne, Lausanne, Switzerland. [2] Laboratory of Photomolecular Science, Institute of Chemical Sciences & Engineering, École Polytechnique Fédérale de Lausanne, Lausanne, Switzerland. [3] Photochemical Dynamics Group, Institute of Chemical Sciences and Engineering, École Polytechnique Fédérale de Lausanne, Lausanne, Switzerland. [4] Present address: H.Glass SA, EPFL Innovation Park, Lausanne, Switzerland. [5] These authors contributed equally: Dan Zhang, Marko Stojanovic, Yameng Ren. ✉email: yameng.ren@epfl.ch; felix.eickemeyer@epfl.ch; anders.hagfeldt@epfl.ch; michael.graetzel@epfl.ch

Since the industrial revolution, tremendous consumption of fossil fuels has led to rapid growth of the global economy and improved quality of life, but at the cost of high risks and impacts of climate change due to the $CO_2$ emission. The Paris agreement, which aims to hold the increase in the global average temperature to well below 2 °C above pre-industrial levels, relies on the development of renewable energy technologies to reduce the consumption of fossil fuels. Photovoltaics (PV) now supply nearly 3% of global electricity reducing $CO_2$ emission and attenuating climate change. Thus, Germany's PV generated 47.5 TWh electric energy in 2019, covering 8.2% of the country's gross electricity demand in the same year and avoiding about 29 million tons of $CO_2$ emission (https://www.ise.fraunhofer.de/en/publications/studies/photovoltaics-report.html).

Compared to the dominant silicon PV, mesoscopic dye-sensitized solar cells (DSCs) have a lower power conversion efficiency (PCE) under standard AM1.5G conditions but industrial applications are emerging for electricity producing glazing and ambient light harvesting to power consumer electronic devices and sensors. The best laboratory scale DSC has achieved a PCE of 13.0% certified by an accredited PV testing laboratory, Fraunhofer Institute for Solar Energy Systems (FhG-ISE) in Freiburg, Germany. The PCE value is presented in NREL's Best Research-Cell Efficiencies Chart. The progress has profited from the development of new sensitizers[1–10], electrolytes[11–15], and device structures[16]. DSCs have advantages of offering bifacial light harvesting, semitransparency, esthetically pleasing appearance, and cost-effective fabrication, making them more suitable for distributing power applications than silicon PV[17]. DSC panels have been deployed as demonstration projects for outdoor applications. They have been installed on the façade of SwissTech Convention Center in 2014[7] and on top of Science Tower of Graz in 2017[18]. Recently, DSCs were found to exhibit an outstanding feature of impressive PCEs over 30% under indoor lighting, superior to amorphous silicon and other thin film semiconductor PV technologies[16]. Thus, they are very promising to serve as power sources to charge portable consumer electronics and sensors. This scenario is being played out, e.g., by the Swedish company Exeger, which is producing flexible DSCs integrated into tablets or earphones to cover their entire energy consumption (https://exeger.com/).

To secure the future successful commercialization of DSCs, further improving their PCE and the durability is warranted. Co-sensitization is one of promising strategies to improve the device performance, especially the short-circuit photocurrent ($J_{sc}$) when the dyes have complementary absorption spectra[9,19]. The concept of co-sensitization goes beyond the development of dyes with high molar extinction coefficients and panchromatic absorption. Recent developments of highly efficient DSCs[16,18,20,21] employ a combination of dyes, one being responsive over a wide spectrum, yielding a large $J_{sc}$, but lower $V_{oc}$, and a co-sensitizer that absorbs mainly blue and yellow light over a narrower spectrum producing a high open-circuit photovoltage ($V_{oc}$) but a low $J_{sc}$. Judicious molecular engineering of the two dyes produces a synergistic effect. Benefiting from the large $J_{sc}$ of the one sensitizer and the high $V_{oc}$ of the other, these photosystems reach a higher PCE and stability than the ones employing either one of dyes[16,18]. In this regard, the development of co-sensitizer with a high $V_{oc}$ offers a promising way to augment the performance of DSCs. Recent studies have shown that some organic dyes together with Cu(II/I) redox mediators are able to achieve a $V_{oc}$ exceeding 1.0 V[15,16,20,22–26], due to the lower energy loss and more positive redox potential of electrolytes employing Cu(II/I) complexes compared to iodide/triiodide or cobalt(III/II) complexes[12,15,27]. Co-sensitized solar cells based on $[Cu^{(II/I)}(tmby)_2][TFSI]_{2/1}$ redox electrolyte (tmby = 4,4′,6,6′-tetramethyl-2,2′-bipyridine;

TFSI = bis(trifluoromethylsulfonyl)imide) have achieved impressive PCEs, both under full sunlight and indoor lighting conditions[16,26].

Here, we show the judiciously designed organic donor-acceptor co-sensitizers, coded MS4 and MS5 (Fig. 1a), featuring the bulky donor N-(2′,4′-bis(dodecyloxy)-[1,1′-biphenyl]-4-yl)-2′,4′-bis(dodecyloxy)-N-phenyl-[1,1′-biphenyl]-4-amine, coined below Hagfeldt donor, and the electron acceptor 4-(benzo[c][1,2,5]thiadiazol-4-yl)benzoic acid (BTBA) for high $V_{oc}$ in DSCs with the $[Cu^{(II/I)}(tmby)_2][TFSI]_{2/1}$ redox electrolyte. Compared to the reference dye NT35 employing the Hagfeldt donor and cyanoacrylic acid (CA)[28] as electron acceptor, MS4 with the same donor but BTBA as acceptor has lower interfacial charge recombination rates and higher $V_{oc}$. By further elongating the end chains of the Hagfeldt donor of MS4 from n-hexyloxy to n-dodecyloxy, the resulting MS5 further suppresses the interfacial charge recombination enabling $V_{oc}$ of 1.24 V, a record among copper-based DSCs. MS5 has a $V_{oc}$ deficit as low as 130 mV and an ideality factor of merely 1.08, which is the lowest value reported for DSCs. The co-sensitization of the high-$V_{oc}$ MS5 with a wide spectral-response dye XY1b shows an impressive PCE of 13.5% under standard AM1.5G sunlight condition and remains stable for 1000 h under light soaking conditions. Remarkably, a MS5 + XY1b co-sensitized DSC with an active area of 2.8 $cm^2$ achieves a record PCE of 34.5% under ambient light at 1000 lux, with a notable $V_{oc}$ of 0.98 V and power output of 109.8 $\mu W\ cm^{-2}$.

## Results

**Synthesis and opto-electronic properties of photosensitizers.** The synthetic routes of MS4 and MS5 are depicted in the Supplementary Information (SI). Figure 1b shows the UV–Vis absorption spectra of these dyes adsorbed on 2.2 μm thick transparent titanium dioxide ($TiO_2$) films in ambient air. The corresponding maximum absorption wavelength is tabulated in Supplementary Table 1. Compared to NT35 with the CA electron acceptor, MS4 with the BTBA electron acceptor shows a 46 nm red shift of maximum absorption wavelength at 468 nm. Substituting the n-hexyloxy chains of Hagfeldt donor in MS4 with n-dodecyloxy groups, yields the MS5 dye, which has a similar absorption spectrum to MS4. Although the maximal molar extinction coefficient of NT35 within 400−500 nm is 4.9-fold and 3.1-fold higher than that of MS4 and MS5 (Supplementary Fig. 1 and Supplementary Table 1), respectively, the maximal absorbance of NT35 on $TiO_2$ is merely 1.5-fold larger than these of both dyes. This result indicates that NT35 has a lower molecular packing density on $TiO_2$ film. The measured dye-loading amount of NT35 in $TiO_2$ film is 2.6 × $10^{-9}$ mol $cm^{-2}$ $\mu m^{-1}$, which is smaller than these of MS4 and MS5 being 5.3 × $10^{-9}$ mol $m^{-2}$ $\mu m^{-1}$ and 6.9 × $10^{-9}$ mol $cm^{-2}$ $\mu m^{-1}$, respectively.

We performed cyclic voltammetry (CV) measurements of the three dyes adsorbed on $TiO_2$ in a three-electrode electrochemical cell to depict their energy level alignments, as shown in Fig. 1c. The CV curves are presented in Supplementary Fig. 2 and the data are tabulated in Supplementary Table 1. The first oxidation potentials ($E_{ox}$) of NT35, MS4, and MS5 on $TiO_2$ film are 1.21, 1.17, and 1.18 V vs. the standard hydrogen electrode (SHE), respectively, being more positive than the redox potential of $[Cu^{(I)}(tmby)_2]TFSI$ (0.87 V vs. SHE)[15], which provides a sufficient driving force for dye regeneration. The zero–zero transition energies ($E_{0−0}$) estimated from the onset absorption wavelength of dyes on $TiO_2$ films (Fig. 1b), are 2.48, 2.29, and 2.28 eV for NT35, MS4, and MS5, respectively. This indicates that the BTBA unit has a stronger electron withdrawing ability than CA, leading to a narrower energy gap for MS4 and MS5 than NT35. The reduction potentials ($E_{red}$) of these dyes, defined as

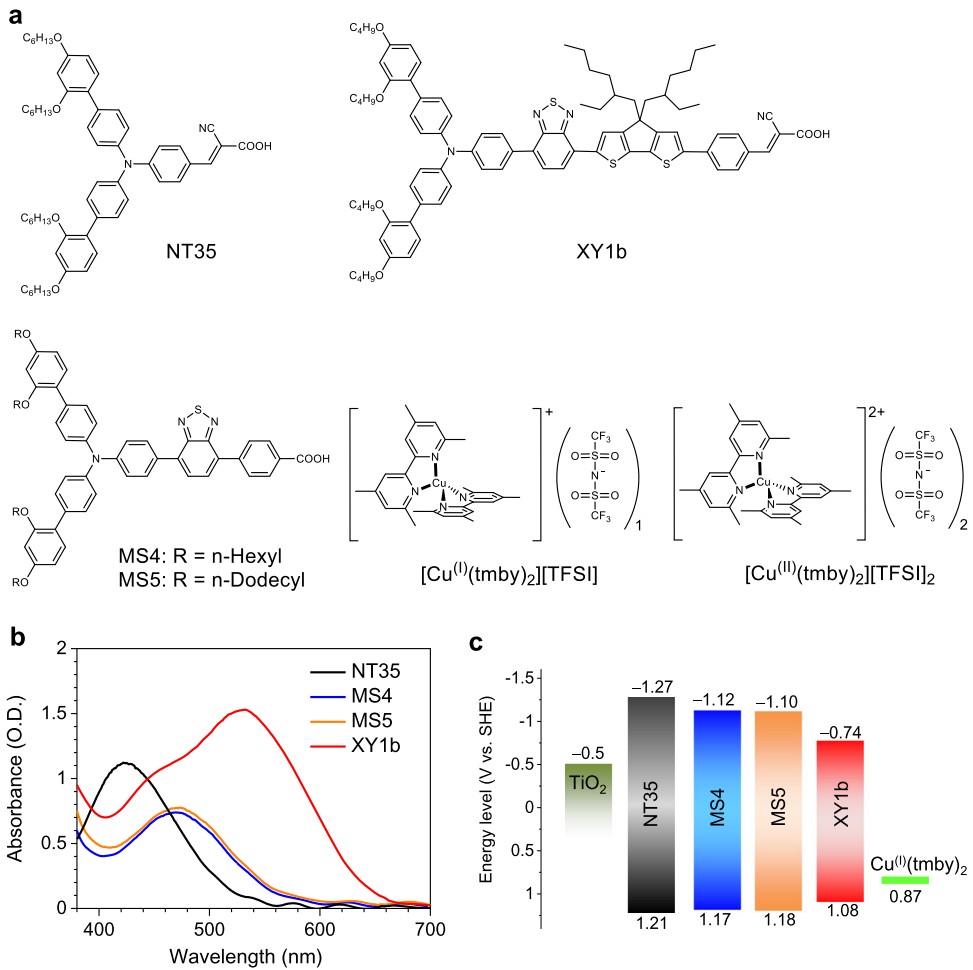

**Fig. 1 Dye molecular structures and their optical and electronic properties. a** Molecular structures of dyes (NT35, MS4, MS5, and XY1b) and copper complex ([Cu(I)(tmby)₂][TFSI] and [Cu(II)(tmby)₂][TFSI]₂, tmby = 4,4′,6,6′-tetramethyl-2,2′-bipyridine; TFSI = bis(trifluoromethylsulfonyl)imide). **b** UV–Vis absorption spectra of NT35, MS4, MS5, and XY1b adsorbed on 2.2 μm thick transparent TiO₂ films. **c** Energy levels diagram of TiO₂, dyes, and [Cu(I)tmby)₂][TFSI].

$E_{red} = E_{ox} − E_{0–0}$, are estimated to be −1.27, −1.12, and −1.10 V vs. SHE for NT35, MS4, and MS5 (Supplementary Table 1), respectively, being more negative than the conduction band edge of TiO₂ electrode (ca. −0.5 V vs. SHE)[29], which ensures sufficient driving force for the electron injection.

**Photovoltaic properties of high $V_{oc}$ DSCs with copper electrolyte**. DSCs based on the three sensitizers in conjunction with the [Cu(II/I)(tmby)₂][TFSI]₂/₁ electrolyte were fabricated and measured according to the description in Methods. The electrolyte for devices performing under AM1.5 G sunlight conditions is composed of 0.2 M [Cu(I)(tmby)₂]TFSI and 0.1 M [Cu(II)(tmby)₂](TFSI)₂ with 0.1 M lithium bis(trifuoromethanesulfonyl)imide (LiTFSI) and 0.6 M N-methylbenzimidazole (NMB) in acetonitrile. Figure 2a shows the incident photon-to-electron conversion efficiencies (IPCEs) as a function of wavelengths of incident monochromatic lights. The IPCE is determined by the efficiencies of light harvesting, the charge injection, the charge regeneration, and the charge collection. MS4 and MS5 have ~50 nm red shift of the onset wavelength of IPCE with respect to NT35, and they show maximum IPCE values over 82% at 510 nm, superior to NT35 (75% at 400 nm). Note that the maximum absorbance of MS4 and MS5 on TiO₂ film is lower than that of NT35 within 400–500 nm (Fig. 1b). The favorable energy level offsets (Fig. 1c)

at the interface of dye/TiO₂ and dye/electrolyte can lead to efficient yields of charge injection and regeneration. The higher IPCE peak of MS4 and MS5 than that of NT35 indicates a more efficient charge collection yield. As evidenced by the following transient photovoltage decay measurement, the devices with MS4 and MS5 have longer electron lifetimes than the one with NT35. The current density–voltage (J−V) curves measured at an irradiance of simulated AM1.5 G sunlight, 100 mW cm⁻², are illustrated in Fig. 2b, and the photovoltaic parameters are tabulated in Table 1. The reference dye NT35 has a $V_{oc}$ of 0.95 V, a $J_{sc}$ of 5.96 mA cm⁻², a fill factor (FF) of 0.791, and a PCE of 4.5%. MS4 featuring a BTBA acceptor instead of CA exhibits a notably higher $V_{oc}$ of 1.17 V, a $J_{sc}$ of 8.86 mA cm⁻², and FF of 0.73, yielding a PCE of 7.6%. By further lengthening the alkyl chains on the Hagfeldt donor from n-hexyloxy (MS4) to n-dodecyloxy (MS5), the PCE increases further to 8.0% ($J_{sc}$ of 8.87 mA cm⁻², and the FF of 0.73). Remarkably MS5 achieves a strikingly high $V_{oc}$ value of 1.24 V, setting a new benchmark for copper-based DSCs. For the given set of components used in this work (dye, oxide, electrolyte), the maximum $V_{oc}$ in a DSC is determined by the difference between quasi-Fermi level of TiO₂, which under intense illumination approaches the energy level of the conduction band edge, and the redox potential of the copper complex[30]. The maximum $V_{oc}$ of [Cu(II/I)(tmby)₂][TFSI]₂/₁ electrolyte-based DSC is estimated to be 1.37 V. Thus, the $V_{oc}$ deficit of the DSC

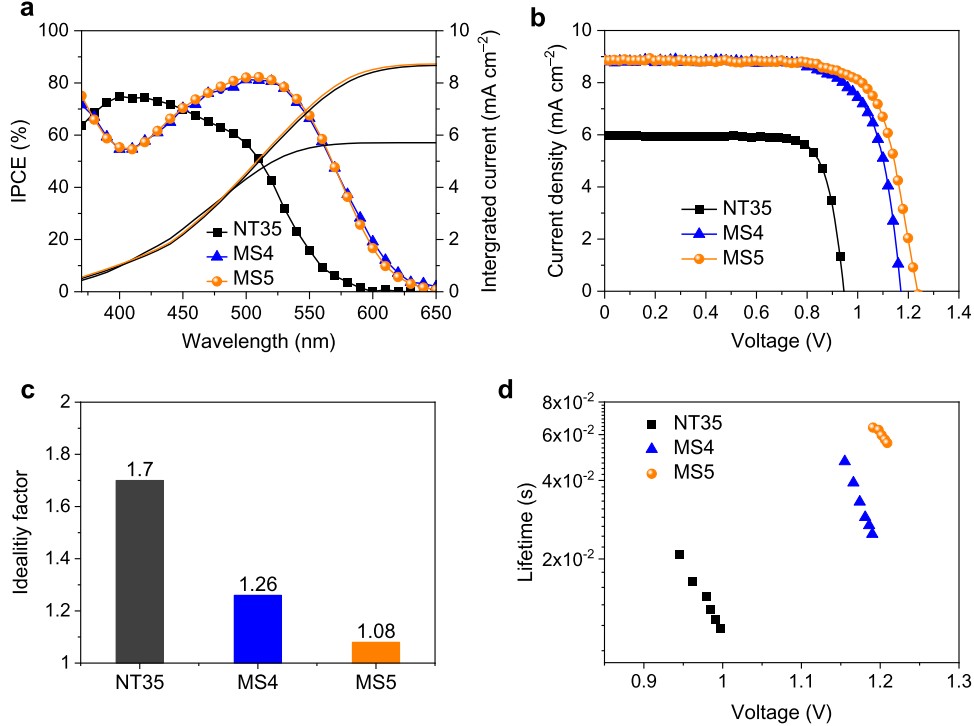

**Fig. 2 Photovoltaic performance and interfacial charge recombination of co-sensitizers. a** Incident photon-to-electron conversion efficiency (IPCE) of the dye-sensitized solar cells (DSCs) based on the co-sensitizers of NT35, MS4, and MS5. The solid lines show the corresponding integrated photocurrent calculated from the IPCE. **b** Current density–voltage curves of the DSCs with NT35, MS4, and MS5 measured under AM1.5G, 100 mW cm$^{-2}$ condition. **c** The ideality factors of the DSCs based on the dyes NT35, MS4, and MS5. **d** Comparison of electron lifetimes measured with the small-pulse transient photovoltage decay method against voltage.

**Table 1 Photovoltaic parameters of DSCs measured under standard AM1.5G sunlight.**

| Dyes | $J_{sc}^{IPCE\,a}$ (mA cm$^{-2}$) | $J_{sc}^{b}$ (mA cm$^{-2}$) | $V_{oc}$ (V) | FF (%) | PCE (%) |
|---|---|---|---|---|---|
| NT35 | 5.71 ± 0.15 | 5.96 ± 0.13 | 0.95 ± 0.004 | 79.1 ± 0.4 | 4.5 ± 0.1 |
| MS4 | 8.69 ± 0.14 | 8.86 ± 0.15 | 1.17 ± 0.002 | 73.0 ± 0.3 | 7.6 ± 0.1 |
| MS5 | 8.73 ± 0.18 | 8.87 ± 0.21 | 1.24 ± 0.003 | 73.3 ± 0.4 | 8.0 ± 0.3 |
| XY1b | 15.02 ± 0.15 | 15.26 ± 0.18 | 1.01 ± 0.003 | 76.3 ± 0.2 | 11.8 ± 0.2 |
| MS5+XY1b | 15.41 ± 0.21 | 15.84 ± 0.24 | 1.05 ± 0.002 | 81.3 ± 0.2 | 13.5 ± 0.2 |

$^{a}J_{sc}^{IPCE}$ values were derived from IPCE spectra. $^{b}J_{sc}$ values were experimental values determined from $J-V$ curves measured under standard AM1.5G, 100 mW cm$^{-2}$ sunlight.

with MS5 and [Cu$^{(II/I)}$(tmby)$_2$][TFSI]$_{2/1}$ is merely 130 mV, indicating a strongly reduced energy loss by interfacial charge recombination.

By fitting the $V_{oc}$ dependance on the light intensity ($\phi$) in Supplementary Fig. 3, the ideality factor ($n$) can be obtained via Eq. (1)

$$n = \frac{q}{k_B T}\frac{dV_{oc}}{d\ln(\phi)} \quad (1)$$

where $q$ is the elementary charge, $k_B$ is Boltzmann's constant, and $\phi$ is incident photon-flux density. $n$ relates to an ideal ($n = 1$) and non-ideal ($n > 1$) diode. For high-quality inorganic semiconductor solar cells, the direct band-to-band radiative charge recombination gives $n \approx 1$. For organic solar cells, $n$ generally departs from unity due to various recombination paths inside band-to-band transitions, such as trap assisted charge recombination. As shown in Fig. 2c, the MS5-based DSC has the lowest $n$ of 1.08 compared to the MS4 and NT35-based counterparts, which is also the lowest value in reported literatures of DSCs. The low value of $n$ of MS5 indicates largely suppressed interfacial charge recombination from trap states.

We further performed charge extraction and transient photovoltage decay measurements to investigate the interfacial energetics and dynamic properties[31,32]. As shown in Supplementary Fig. 4, at a certain photovoltage an almost invariable amount of charge can be extracted from TiO$_2$ films for all cells, implying that the change of dyes may not alter the conduction band edge and electron trap states of TiO$_2$. At a given density of extracted charge, the electron lifetimes of MS4-based devices are about two orders of magnitude longer than these of NT35-based counterpart (Fig. 2d). As mentioned above, MS4 has a higher dye-loading on TiO$_2$ film compared to NT35 that potentially depresses interfacial charge recombination. Compared to MS4, the electron lifetimes in the MS5 are about 3-times longer, partly due to the slightly higher dye-loading amount of MS5 and longer alkyloxy chain on the Hagfeldt donor to hinder the approach of Cu(II) to the TiO$_2$ surface to recombine.

**Highly efficient co-sensitized solar cells under AM1.5G sunlight.** Benefiting from its high $V_{oc}$ and strong capacity of retarding interfacial charge recombination, MS5 presents promising features to serve as a co-sensitizer to realize highly efficient

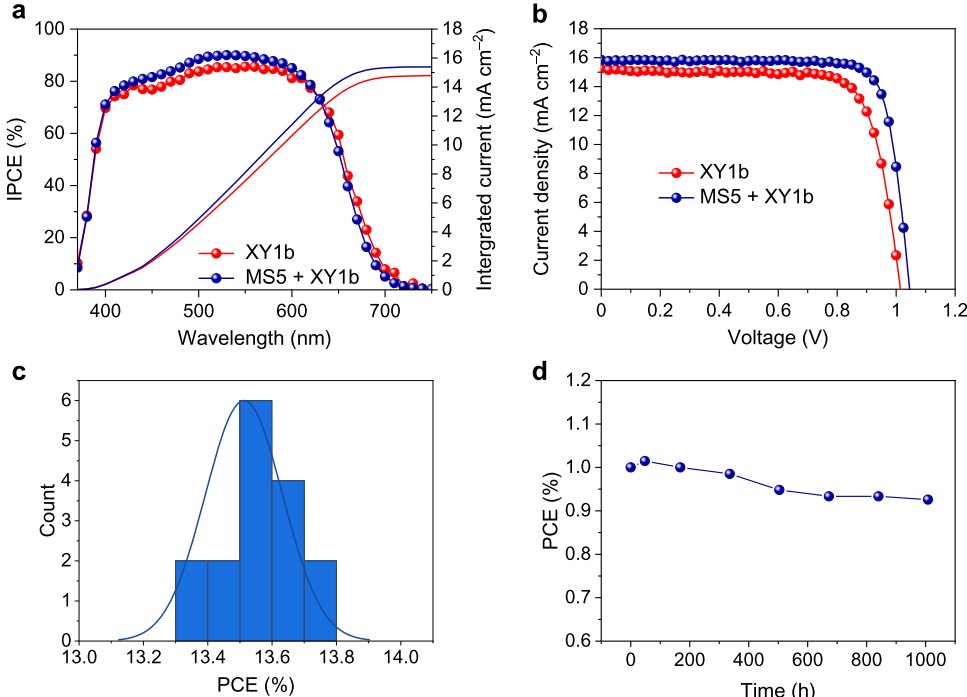

**Fig. 3 Photovoltaic performance of solar cells based on XY1b and MS5+XY1b. a** Incident photon-to-electron conversion efficiency (IPCE) of the dye-sensitized solar cells (DSCs) based on XY1b and the co-sensitization of MS5+XY1b. The solid lines show the corresponding integrated photocurrent calculated from the IPCE. **b** Current density–voltage curves of the DSCs with XY1b and the co-sensitization of MS5 + XY1b measured under AM1.5G, 100 mW cm$^{-2}$ condition. **c** Histogram of power conversion efficiency (PCE) of the DSCs based on co-sensitization of MS5+XY1b (16 samples). **d** Evolution of PCE of the DSCs based on MS5+XY1b measured under AM1.5G sunlight (100 mW cm$^{-2}$) during continuous light soaking at 45 °C for 1000 h.

DSCs. Herein, we employ MS5 as a co-sensitizer together with the previously reported wide spectral-response dye XY1b[16] (Fig. 1a). The absorption spectrum of XY1b adsorbed on a mesoscopic TiO$_2$ film is shown in Fig. 1b. XY1b has a strong absorption peak at 531 nm, which is red shifted by 63 nm compared to the absorption peak of MS5 within 400−700 nm. The IPCE spectrum of the MS5 + XY1b-based DSC attains slightly higher values than that of XY1b-based counterpart, as shown in Fig. 3a. Figure 3b shows the J−V curves measured under simulated AM1.5 G 100 mW cm$^{-2}$ sunlight conditions. The control DSC based on XY1b alone shows a PCE of 11.8%, with a $J_{sc}$ of 15.26 mA cm$^{-2}$, a $V_{oc}$ of 1.01 V, and a FF of 0.763. The performance is similar to our previous result[16]. By contrast, the MS5 + XY1b co-sensitized solar cell exhibits an enhanced $V_{oc}$ of 1.05 V, a slightly higher $J_{sc}$ of 15.84 mA cm$^{-2}$, which is in good accordance with the integrated current density calculated from IPCE (Fig. 3a and Table 1), and a remarkable FF of 0.813, overall delivering a benchmark PCE of 13.5% (Supplementary Table 2). The PCE histogram of the MS5 + XY1b cells (Fig. 3c) demonstrates the good reproducibility of our results. As shown in Fig. 3d, MS5 + XY1b-based DSC exhibits excellent photostability, maintaining 93% of its initial value during light soaking at full solar intensity (100 mW cm$^{-2}$) at 45 °C for 1000 h. The detailed evolutions of photovoltaic metrics are presented in Supplementary Fig. 5.

**Analysis of the loss of photovoltaic performances under AM1.5G sunlight**. To unravel the fundamental reasons for the substantial improvement of the performance by co-sensitization, we studied the major PCE loss mechanisms of the XY1b and MS5 + XY1b-based DSCs by analyzing the J−V curves (Supplementary Fig. 6) following a method inspired by Guillemoles et al. and Stolterfoht et al.[33,34]. The performance losses of $J_{sc}$, $V_{oc}$, and FF are summarized in Fig. 4a. For the determination of these losses we start with the calculation of a J−V curve for the

DSC-equivalent of the Shockley–Queisser limit, which we call quasi-Shockley–Queisser (qSQ) limit in the following (see Supplementary Fig. 6, red curves). For the qSQ limit of $V_{oc}$ we take the difference between the Fermi level of TiO$_2$ and the redox potential of the copper complex yielding $V_{oc,qSQ} = 1.37$ V, which represents the maximum $V_{oc}$ that can be achieved with this DSC as mentioned above. The qSQ limit of $J_{sc}$ is calculated assuming complete solar light absorption, i.e., 100% IPCE, above the absorption onset energy $E_o$. The latter is determined from the inflection point of the IPCE spectra[35]. The qSQ J−V curve is then calculated from $V_{oc,qSQ}$ and $J_{sc,qSQ}$ using Eq. (2)[34]

$$J(V) = J_{0,qSQ}\left(e^{aV/nk_BT-1}\right) - J_{sc,qSQ} \qquad (2)$$

with the dark emission current $J_{0,qSQ}$ being determined from the relation $0 = J_{0,qSQ}\left(e^{qV_{oc,qSQ}/nk_BT} - 1\right) - J_{sc,qSQ}$, where $q$ is the elementary charge, $k_B$ the Boltzmann constant and $T = 298$ K the cell temperature (see Supplementary Fig. 6 and Supplementary Table 3). The ideality factor $n$ is taken as 1.

As shown in Fig. 4a, the $J_{sc}$ loss of MS5 + XY1b (6%) is only half as big as the value for XY1b (12.4%). Although the MS5 + XY1b-based DSC has a slightly blue shifted onset wavelength for the photocurrent response and light-harvesting efficiency spectrum (Supplementary Fig. 7) with respect to XY1b, the higher IPCE for MS5 + XY1b contribute to its smaller $J_{sc}$ losses. To reveal the reasons for the smaller $J_{sc}$ loss in the co-sensitized solar cell, we employed nano-second laser flash photolysis technique[36,37] to study the dual-path charge transfer kinetics of the oxidized dye molecules (D$^+$) being reduced by either Cu(I) ions in the electrolyte or by recapturing the injected electrons in TiO$_2$. As shown in Fig. 4b, when the dye-sensitized TiO$_2$ films are in contact with an inert electrolyte consisting of 0.1 M LiTFSI and 0.6 M NMB in acetonitrile, the transient absorption decay signals reflecting the back-electron-transfer

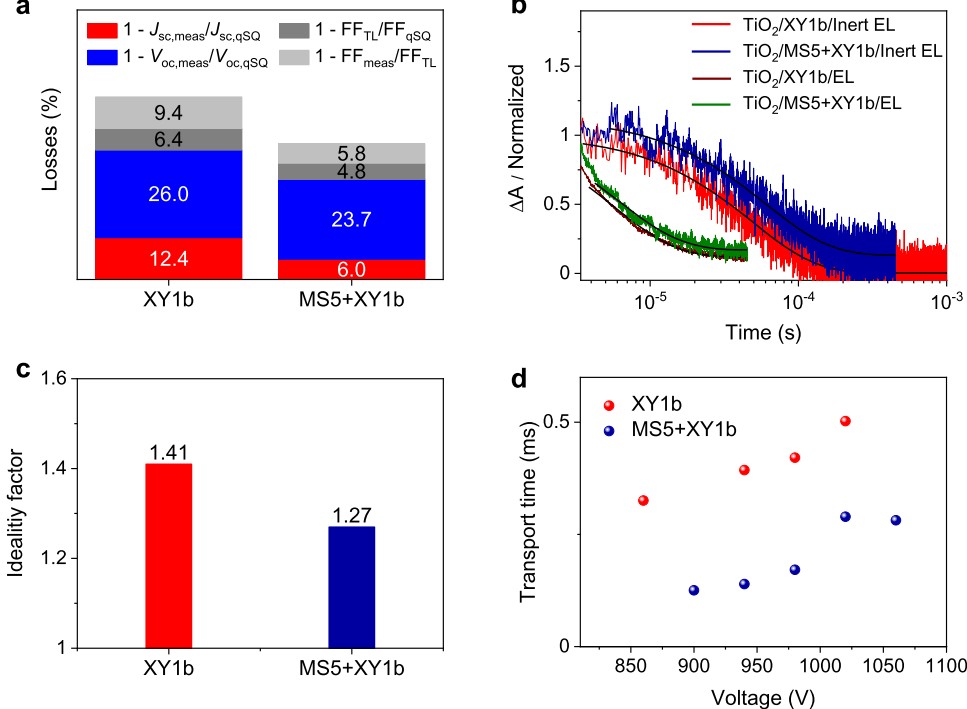

**Fig. 4 The loss mechanism of photovoltaic parameters of XY1b and MS5 + XY1b. a** Photovoltaic performance losses due to non-ideal absorption (red), non-radiative recombination (blue and dark gray) and charge transport (light gray). The subscripts stand for measured current density–voltage (J−V) curve (meas), quasi Shockley–Queisser limit (qSQ), and transport limit (TL). $V_{oc}$ and FF stand for open-circuit photovoltage and fill factor, respectively. **b** Nanosecond Flash photolysis measurements of dyes-sensitized $TiO_2$ films immersed in a $[Cu^{(II/I)}(tmby)_2][TFSI]_{2/1}$ redox electrolyte (EL) or an inert electrolyte (tmby = 4,4′,6,6′-tetramethyl-2,2′-bipyridine; TFSI = bis(trifluoromethylsulfonyl)imide). The solid lines are monoexponential fittings. Pump wavelength: 520 nm; probe wavelength, 815 nm. **c** The ideality factors of the DSCs based on the dyes XY1b and MS5 + XY1b. **d** The transport time as a function of applied voltage.

from the conduction band of $TiO_2$ to $D^+$ have a time constant of 49.0 μs and 61.0 μs for XY1b and MS5 + XY1b, respectively. When the dye-sensitized $TiO_2$ films are contacted by the $[Cu^{(II/I)}(tmby)_2][TFSI]_{2/1}$ electrolyte, the decays, reflecting the regeneration of $D^+$ by $[Cu^{(I)}(tmby)_2]TFSI$, display lifetimes that are one order of magnitude shorter (5.7 and 6.0 μs for XY1b and MS5 + XY1b, respectively). Overall, the calculated dye regeneration efficiencies are around 86.4% and 91.0% for XY1b and MS5 + XY1b, respectively, which partly explains the higher IPCE plateaus and lower $J_{sc}$ losses of co-sensitized solar cells.

The $V_{oc}$ losses are 26.0% and 23.7% for XY1b and MS5 + XY1b, respectively. The $V_{oc}$ losses are due to residual non-radiative recombination at the $TiO_2$/electrolyte interface. The charge extraction and transient photovoltage decay measurements show that the MS5 + XY1b impregnated $TiO_2$ film has a threefold longer electron lifetime than that sensitized by XY1b alone (Supplementary Fig. 8), which proves that co-sensitization is effective in retarding the interfacial charge recombination. Note also the higher dye-loading amounting to $2.70 × 10^{-8}$ mol $cm^{-2}$ $μm^{-1}$ for MS5 + XY1b (XY1b: $1.77 × 10^{-8}$ mol $cm^{-2}$ $μm^{-1}$ and MS5: $0.93 × 10^{-8}$ mol $cm^{-2}$ $μm^{-1}$) compared to only $1.97 × 10^{-8}$ mol $cm^{-2}$ $μm^{-1}$ for XY1b alone, which retards the recombination of conduction band electrons in $TiO_2$ with copper(II) ions in the electrolyte.

We evaluated the contribution from non-radiative recombination by calculating a J−V curve from the diode equation using the measured $J_{sc}$, $V_{oc}$, and ideality factor n. We determined n from the light intensity dependent $V_{oc}$ measurement of the DSCs (Supplementary Fig. 9). As shown in Fig. 4c, MS5 + XY1b sensitized DSC has a smaller n of 1.27 than the value n = 1.41 derived for XY1b. Since this calculated J−V curve (blue curves in

Supplementary Fig. 6) takes into account non-radiative recombination losses but neglects losses stemming from non-ideal charge transport, the FF transport limit ($FF_{TL}$) of this transport limit J−V curve indicates the contribution from non-radiative recombination. The MS5 + XY1b cell has a substantially smaller loss of $FF_{TL}$ (4.8%) than the XY1b cell ($FF_{TL}$ = 6.4%), indicating a suppression of non-radiative recombination in the co-sensitized system. In a real device, charge transport losses manifest themselves in a further reduction in FF.

The FF losses of the measured J−V curve with respect to $FF_{TL}$ are 9.4% and 5.8% for XY1b and MS5 + XY1b-based DSCs, respectively, indicating a smaller transport loss in co-sensitized system. To scrutinize the reason for the difference of the transport losses between XY1b and MS5 + XY1b, electrochemical impedance spectroscopy (EIS) measurements of devices under white light LED illumination with 100 mW $cm^{-2}$ intensity were performed. The measured spectra were fitted to the data by employing the ZView software and the transmission-line model[38]. As shown in Fig. 4d, the transport time of the MS5 + XY1b-based DSC is shorter than that of XY1b alone based counterpart, which indicates a faster electron transport through the mesoscopic $TiO_2$ layer to the external circuit and more efficient charge collection, contributing to its smaller transport loss. In addition, the smaller Warburg diffusion resistance for the MS5 + XY1b-based DSC with respect to that of XY1b counterpart (Supplementary Fig. 10) contributes to its smaller transport loss. The detailed molecular origin of this observation is not clear and beyond the scope of this work.

**Highly efficient co-sensitized solar under ambient lighting**. The DSC is a unique photovoltaic technique that has achieved

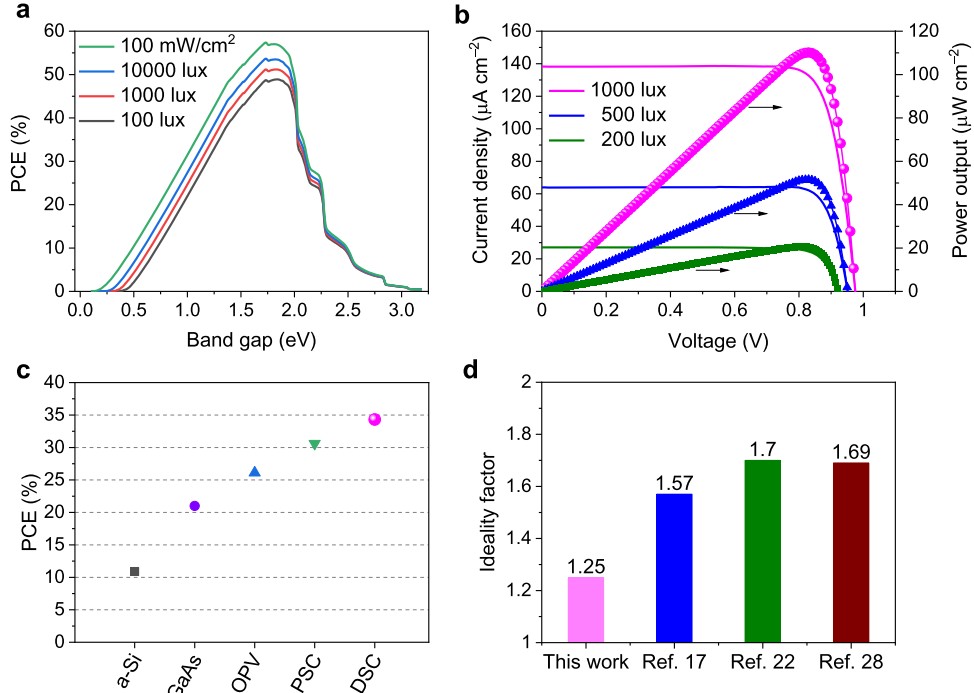

**Fig. 5 The theoretical maximal PCE and the photovoltaic performance of MS5 + XY1b-based DSC under indoor lighting. a** Band gap dependent simulations of the theoretical power conversion efficiency (PCE) limitations (Shockley–Queisser limit) for different intensities of the model Osram 930 Warm White fluorescent tube light. **b** Current density–voltage curves and power output of the dye-sensitized solar cell (DSC) with a photoactive area of 2.80 cm$^2$ under different ambient light intensities. **c** Summary of the best indoor PCEs of various types of solar cells with the photoactive area of at least 1 cm$^2$, including amorphous silicon (a-Si), GaAs, organic polymers (OPV), metal halide perovskite (PSC), and DSC. **d** Summarized the ideality factors of representative publications on DSCs under ambient light.

impressive PCE under ambient light. Different from direct solar radiation, only a relatively small number of electrons can be generated under the lower incident power of ambient light compared to full sun illumination. Therefore, the suppression of recombination processes plays a crucial role in the DSC performance under diffuse light. The above results imply that the DSC based on MS5 + XY1b can suppress the charge recombination very well, and the ideality factor close to one implies a low level of non-radiative losses that there is a small $V_{oc}$ decrement with decreasing light intensity. In this regard, a DSC based on MS5 + XY1b with an active area of 2.8 cm$^2$ was fabricated to investigate its performance under ambient lighting using a standard Osram 930 Warm White fluorescent tube as light source. Depending on the energy gap of a light absorbing material under different intensities of the model Osram 930 Warm White fluorescent tube light, the theoretical PCE limitations (Shockley–Queisser limit) are calculated with etaOpt software[39], as shown in Fig. 5a. Owing to the narrow band distribution of the spectra (Supplementary Fig. 11), the energy gap of 1.73 eV yields a theoretical maximum PCE of 49%, 51%, 54%, and 57% at the light intensity of 100 lux, 1000 lux, 10000 lux, and 314,465 lux (100 mW cm$^{-2}$) under Osram 930 Warm White fluorescent tube condition, respectively. Note that the energy gap of MS5 + XY1b system is estimated to be 1.72 eV from the onset of the IPCE spectrum, which matches well the optimal band gap for solar cells working under the Osram 930 Warm White fluorescent tube indoor lighting.

The DSCs tested under ambient light employed an electrolyte with lower concentrations of Cu(II) complexes to support the lower photocurrents generated under these conditions by the photovoltaic[17]. As shown in Supplementary Table 4, reducing the concentration of Cu(II) can suppress interfacial charge recombination and improve in particular the $V_{oc}$ and overall device performance, under ambient light. We further found that

**Table 2 Photovoltaic parameters of the DSC based on MS5 + XY1b with photoactive area of 2.80 cm$^2$ (length: 40 mm; width: 7 mm) under different light intensities from an Osram 930 Warm White fluorescent light.**

| Cell performance | $P_{in}$ of Osram 930 Warm White Light (lux or µW cm$^{-2}$) | | |
|---|---|---|---|
| | **1000 (318.2)** | **500 (159.1)** | **200 (63.6)** |
| $P_{max}$ (µW) | 307.4 | 143.9 | 57.7 |
| PCE (%) | 34.5 | 32.3 | 32.4 |
| $V_{oc}$ (V) | 0.98 | 0.95 | 0.92 |
| $I_{sc}$ (mA) | 0.387 | 0.179 | 0.076 |
| FF | 0.815 | 0.845 | 0.826 |

lowering the concentration of Cu(I) also benefited the PCE. Devices tested under the ambient light employed an electrolyte composed of 0.1 M [Cu$^{(I)}$(tmby)$_2$]TFSI and 0.04 M [Cu$^{(II)}$(tmby)$_2$](TFSI)$_2$ complexes together with 0.1 M LiTFSI and 0.6 M NMB in acetonitrile. The $J-V$ curves of DSCs were recorded at 1000, 500, and 200 lux, as shown in Fig. 5b. The photovoltaic parameters are tabulated in Table 2. At 1000 lux, the DSC achieves an impressively high $V_{oc}$ of 0.98 V, exceeding that of all previously reported devices. Along with the short-circuit photocurrent of 387 µA ($J_{sc}$ = 138.2 µA cm$^{-2}$) and FF of 0.815, the device produces a maximal power of 307.4 µW ($P_{max}$ = 109.8 µW cm$^{-2}$) that corresponds to an unprecedented PCE of 34.5%. Furthermore, DSCs with larger areas of 8.0 cm$^2$ and 20.25 cm$^2$ achieved PCEs of 31%[26] and 26%[16] at 1000 lux, respectively. These results place DSCs in a leading position for ambient light-harvesting applications, surpassing the performance of other types of solar cells made of amorphous silicon

(a-Si)[40], GaAs[20], organic polymers (OPV)[41], and metal halide perovskite (PSC)[42] (Fig. 5c and Supplementary Table 5). At 500 and 200 lux, the $V_{oc}$ of the DSC remains over 0.92 V and delivers PCEs of 32−33% (Table 2). As shown in Fig. 5d, the ideality factor $n$ of this device is 1.25 under dim light, lower than that of the previous benchmark DSCs[16,20,26] (Supplementary Fig. 12 and Supplementary Table 6). Remarkably, the DSC retain well its initial PCE after 500 h under continuous illumination under indoor light (Supplementary Fig. 13), reflecting excellent stability.

## Discussion

In conclusion, we designed and synthesized two simple high-$V_{oc}$ organic dyes (MS4 and MS5) through judiciously tailoring the electron acceptor and Hagfeldt donor for copper-based DSC. The dye MS5, characteristic of Hagfeldt donor with $n$-dodecyl chains and the electron acceptor BTBA, yields the highest $V_{oc}$ of 1.24 V with the $V_{oc}$ deficit as low as 130 mV and ideality factor of 1.08, benefited from a retarded interfacial charge recombination. The co-sensitization of MS5 and XY1b achieves highly efficient and stable DSCs with a remarkable high PCE of 13.5% under standard AM1.5G sunlight condition. Strikingly, under standard Osram 930 Warm White fluorescent tube light at 1000 lux intensity, the DSC with the active area of 2.8 cm² presented an impressive PCE of 34.5% with a distinctive $V_{oc}$ of 0.98 V and power outputs of 109.8 μW cm⁻². Such high PCEs stems from a very low ideality factor, mainly because of the reduced interfacial charge recombination. Our work highlights the importance of judicious molecular engineering of high-$V_{oc}$ co-sensitizers to improve photovoltaic performance of DSCs based on copper redox electrolyte.

## Methods

**Materials**. Acetonitrile (ABCR), chloroform (CF) (ABCR), *tert*-butanol (Sigma-Aldrich), ethanol (EtOH) (Acros), *N*-methylbenzimidazole (Sigma-Aldrich), chenodeoxycholic acid (Sigma-Aldrich), lithium bis(trifluoromethanesulfonyl)imide (LiTFSI) (Sigma-Aldrich). The powders of [Cu(I)(tmby)₂][TFSI] and [Cu(II)(tmby)₂][TFSI]₂ were purchased from Dyenamo AB, and used as received without further purification. The powders of XY1b dye molecules were purchased from Dyenamo AB, and further purified before the use. The dye NT35 was synthesized according to the previous[28]. The photosensitizing dye MS4 and MS5 were prepared according to the synthetic routes in the Supplementary Fig. 14.

**Fabrication of solar cells**. The mesoporous TiO₂ electrodes were fabricated following the literature procedure[16]. The Fluorine doped tin oxide (FTO) glass (Nippon Sheet Glass, NSG, 10 ohms sheet resistance) was thoroughly cleaned with water, acetone, and ethanol in sequence. Then the substrates were treated by 50 mM TiCl₄ aqueous solution at 70 °C for 50 min before a 15 min ultraviolet/O₃. The TiO₂ pastes of 18 NR-T or 30 NR-D (Greatcell Solar Limited) and the light-scattering TiO₂ particles (Greatcell Solar Limited, WER2-0) were sequentially deposited on the above FTO glass by screen-printing technology. The TiO₂ films were patterned in round spots with the area of 0.28 cm² or rectangular shapes (40 × 7 mm). The mesoporous TiO₂ films composed of ~4.0 μm transparent layer and ~4.0 μm light-scattering layer were obtained after the films were sintered in dry air flow and gradually cooled down to room temperature. The mesoporous TiO₂ films for ambient light devices were treated in TiCl₄ (40 mM) for 40 min in an oven at 70 °C. The mesoporous TiO₂ films for 1 sun devices were without TiCl₄ solutions post-treatment. The mesoporous TiO₂ electrodes were stained by immersing them into dye solutions at room temperature for 14 h before sintered at 500 °C in air for 30 min and cooled down to 80 °C. The dye solutions of NT35, MS4, and MS5 was prepared by dissolving 0.1 mM corresponding dye in acetonitrile/*tert*-butanol (*v/v*, 1/1). The dye solution of XY1b was made by dissolving 0.1 mM XY1b and 2.5 mM chenodeoxycholic acid in CF/EtOH (*v/v*, 1/9). The solution for co-sensitization of MS5/XY1b was prepared by dissolving 0.05 mM MS5, 0.1 mM XY1b, and 0.5 mM chenodeoxycholic acid in CF/EtOH (*v/v*, 1/9). The counter electrode (the PEDOT film coated FTO glass) was prepared using the electrical deposition technique following the literature procedure[16]. Both electrodes were pressed together mechanically without spacer and further sealed with UV light curing glue (ThreeBond 3035B), which is quickly solidified by UV light from a UV curing machine (TEKLITE). The electrolyte was injected into the sealed electrodes through a predrilled hole on the counter electrode to complete the fabrication of the sandwich-type DSC. The hole was sealed with UV light curing glue. All cells for

characterizations were sealed. The copper-based electrolyte consisting of 0.2 M [Cu(I)(tmby)₂]TFSI and 0.1 M [Cu(II)(tmby)₂](TFSI)₂ complexes with 0.1 M LiTFSI and 0.6 M NMB in acetonitrile was used to obtain high efficiency under one sun. The copper-based electrolyte consisting of 0.1 M [Cu(I)(tmby)₂]TFSI and 0.04 M [Cu(II)(tmby)₂](TFSI)₂ complexes with 0.1 M LiTFSI and 0.6 M NMB in acetonitrile was used to obtain high efficiency under ambient light.

**Characterization of solar cells**. During the IPCE and J−V measurements, the DSCs with a photoactive area of 0.28 cm² were masked with an aperture area of 0.158 cm². The simulated solar light was provided by a 300 W Xenon light source from Oriel, which is equipped with a SchottK113 Tempax sunlight filter (Praezisions Glas & OptikGmbH) to match the emission spectrum of the lamp to the AM1.5G standard. The light intensity was determined using a calibrated Si reference diode equipped with an infrared cutoff filter (KG-3, Schott). The attenuated light intensities of ~50 and ~10 mW cm⁻² were obtained by using the metal mesh. The DSCs with the photoactive areas of 2.80 cm² were masked with rectangular areas of 3.80 cm² and the artificial indoor light was provided by the OSRAM 930 Warm White tube light. The light intensities were calibrated by the light meter (TES-1334, TES). The J−V curves were recorded by a Keithley 2400 source meter. The voltage scan rate was set to 125 mV s⁻¹. IPCE was recorded with a commercial apparatus (Aekeo-Ariadne, Cicci Research s.r.l.) based on a 300-W Xenon lamp. During the measurement, the solar cell was illuminated under the white light at 10 mW cm⁻² supplied by an array of white LED. For light soaking ageing test at intensity of 100 mW cm⁻² at 45 °C, the solar cells were stressed under open-circuit conditions. For stability test, the devices with UV light curing glue sealant were further sealed with 3 M™ Scotch Weld™ Epoxy Adhesive DP460 to prevent the electrolyte from leaking and block molecules such as water and oxygen from penetrating into the devices.

**Electrochemical and UV–Vis absorption spectra characterizations**. The electrochemical characterization of metal oxide surface-attached dyes was performed on a BioLogic SP300 potentiostats in a classical three-electrode configuration using electrochemical cyclic voltammetry (CV) and differential pulse voltammetry (DPV) procedures. The dye adsorbed on 2.2 μm thick transparent mesoscopic TiO₂ film was used as working electrode, a graphite rod was used as counter electrode, and a silver wire covered with silver chloride (Ag/AgCl) immersed in a 0.1 M LiTFSI solution in acetonitrile was used as a quasi-reference electrode (QRE) in acetonitrile in a sealed bridge tube. The supporting electrolyte was 0.1 M LiTFSI in dry acetonitrile. The QRE was calibrated against the ferrocene/ferrocenium redox system by cyclic voltammetry. A Perkin-Elmer Lambda 950 spectrophotometer was used to record UV–Vis absorption spectra. The UV–Vis absorption spectra of dyes adsorbed on 2.2 μm thick transparent TiO₂ films were measured in ambient air.

**Transient photovoltage decay and charge extraction measurements**. Electron lifetime and extracted charge measurements were performed using the Dyenamo toolbox using a white LED (Luxeon Star 1 W) as light source. Voltage traces and current traces were recorded with a 16-bit resolution digital acquisition board (National Instruments), then lifetimes and extracted charges were determined by monitoring photovoltage transients at different light intensities upon applying a small square wave modulation to the base light intensity. The photocurrent and photovoltage responses were fitted with using first-order kinetics to obtain time constants.

**Nano-second flash photolysis measurement**. The samples were excited at λ = 520 nm by 10 ns-duration laser pulses produced by a tunable optical parametric oscillator (OPO-355, GWU) pumped by a frequency-tripled (355 nm), Q-switched Nd:YAG laser (Surelite, Continuum, 20 Hz repetition rate). The excitation light fluence was reduced by use of various neutral density filters to 0.6 mJ cm⁻² per pulse (for samples with redox-active electrolyte) and 2.2 μJ cm⁻² per pulse (for samples without redox-active electrolyte). The time evolution of the absorbance change of the sample was probed at λ = 815 nm by passing a cw light beam produced by a Xe arc lamp through various IR and cutoff filters and a first monochromator before reaching the sample at an angle of 60°. The probe beam was finally collected by a second grating monochromator (CS260, Newport) and detected by a fast Si avalanche photodiode (APD410A/M, Thorlab). The transient voltage signal was recorded by a fast oscilloscope (DPO 7104C, Tektronix) and averaged over 5000 laser shots. The data were extracted with a MATLAB routine and fitted using a first-order decay Eq. (3)

$$f(x) = y0 + A*e^{-x/\tau} \tag{3}$$

**Electrochemical impedance measurements**. Impedance measurements were performed using a BioLogic SP300 potentiostat, over a frequency range from 1 MHz down to 0.1 Hz at bias potentials between 0 and 1.06 V (with a 40-mV sinusoidal AC perturbation). All measurements were done at 25 °C. The resulting impedance spectra were analyzed with Z-view software (v2.8b, Scribner Associates Inc.).

**Reporting summary**. Further information on research design is available in the Nature Research Reporting Summary linked to this article.

## Data availability

All relevant data in this study are available from the corresponding authors upon request.

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

## Acknowledgements

Y.R., S.M.Z., and M.G. acknowledge the financial support from the European Union's Horizon 2020 research and innovation program under grant agreement No. 826013. D.Z. and A.H. is grateful for the financial support of the Swiss National Science Foundation under contract SNSF 200020_185041. Authors acknowledge B. Delayre for his help with IR and melting point measurements.

## Author contributions

M.G. and A.H. supervised the study. Y.R. conceived the idea and designed the experiments. M.S. designed MS4 and MS5 sensitizers and realized their synthesis, and characterized the dyes. D.Z. and Y.R. fabricated and characterized the solar cells. D.Z. performed photostability test. M.S. measured the UV–Vis spectroscopies. D.Z., M.S., Y.R., and N.V. performed electrochemical characterization of metal oxide surface-attached dyes and carried out the data analysis. F.T.E. calculated theoretical efficiency limitations under indoor light and calculated the photovoltaic performance losses. E.C.S. and J.-E.M. measured the nano flash photolysis and carried out data analysis. D.Z., Y.R., M.S. Y.C., and F.T.E. discussed the results and assembled the Figures. D.Z. and Y.R. wrote the initial manuscript with input from all authors while M.G. wrote the final version of the manuscript. Y.R. and M.G. assumed all correspondence with the editor and reviewers. S.M.Z. and M.G. coordinated the work.

## Competing interests

The authors declare no competing interests.
