## [Peer Review File · Nature Communications]

REVIEWER COMMENTS

Reviewer #1 (Remarks to the Author):

This paper describes the synthesis, optical, electrochemical, and photovoltaic properties of novel push-pull dyes, MS4 and MS5 for copper(I/II) redox couple-based dye-sensitized solar cells (DSCs). The authors achieved a highest open-circuit voltage (V_{oc}) of 1.24 V using MS4 dye. Moreover, a combination of MS5 and XY1b dyes led to a remarkable power conversion efficiency (PCE) of 13.5% under AM1.5 condition and a PCE of 34.5% under ambient light condition. Such photovoltaic performances are very impressive and may attract considerable interest from researchers in the field of dye-sensitized solar cells (DSSCs). However, I have to raise a couple of concerns that should be adequately addressed before the publication.

1) I cannot find the IR and melting point of the new compounds in the supporting information. The data of NMR, high-resolution mass spectrum, IR, and melting point are necessary to identify the structures of the new compounds.

2) The absorption peaks of MS4 and MS5 on TiO_2 appear at 470 nm in Fig. 1, whereas the corresponding peaks of IPCE values are seen at 520 nm. The discrepancy should be rationalized.

3) The oxidation potentials were determined by cyclic voltammetry (Supplementary Fig. 2). The values listed in lines 107-110 do not seem to match those taken from the Fig. 2. Moreover, the electrochemical processes are irreversible or quasi-irreversible. They should use differential pulse voltammetry to determine the oxidation potentials more accurately.

Reviewer #2 (Remarks to the Author):

In this study, two high- V_{oc} organic dyes of MS4 and MS5 have been designed and synthesized. The highest V_{oc} of 1.24V was obtained by MS5 and co-sensitization with XY1b achieved an efficiency of 13.5% under standard AM1.5G sunlight condition. Also, the DSC with the active area of 2.8 cm^2 achieved a record PCE of 34.5% under standard Osram 930 Warm White fluorescent tube light at 1,000 lux intensity. This is an extremely good work in DSSCs. Thus, I recommend it for publication in Nature Communications after a minor revision. Some issues need to be addressed:

Q1. In Supplementary Fig. 1, MS4 and MS5 dyes displayed their highest absorption at around 330 nm. Thus, in Supplementary Table 1, these values around their λ_{max} (330nm) should be added.

Q2. In the third paragraph of 'Results and discussion', the authors addressed that the higher IPCE of MS4 and MS5 indicates a more efficient charge collection yield. However, as far as we can see, the value of IPCE can be also influenced by the light harvest efficiency and charge injection efficiency, as well as the charge regeneration efficiency. Please clarify it more clearly.

Q3. "MS5+XY1b"-based DSC exhibits excellent photostability and maintained 93% of its initial value during light soaking at full solar intensity for 1,000 hours. How could the device be stable up to 1000 hours with only a little PCE loss? Please provide how to encapsulate the device in more details.

Q4. The dye-loading amounting is $2.70 \times 10^{-8} \text{ mol cm}^{-2} \mu\text{m}^{-1}$ for "MS5+XY1b". Is it possible to also show the dye-loading amount of each dye of MS5 and XY1b, respectively?

Q5. We noticed that the authors used different concentrations of copper electrolyte for application in AM1.5G sunlight and ambient light. Thus, how will concentration change affect the efficiency of DSSCs, especially in ambient light?

Q6. According to the authors in line number 63, the authors addressed "Benefiting from the high V_{oc} of one dye and the high J_{sc} of the other, co-sensitized solar cell can reach a higher PCE and stability than the one employing either one of dyes". It seems that the band gap of dye can determine the V_{oc} . Generally speaking, co-adsorption is effective when two dyes have supplementary absorption spectra, leading to a high J_{sc} . Please clarify them more clearly.

Q7. According to the equation 1 and n value of "MS5+XY1b" in Supplementary Fig. 9, it was calculated that the Voc of "MS5+XY1b" should not be higher than 0.88 V at 1000 lux ($318.2 \mu\text{W cm}^{-2}$), as Voc shows a linear relationship with $\ln I$. But, in this report, the authors found the Voc of MS5+XY1b around 0.98 V. Furthermore, the lowest value of light intensity in case of MS5+XY1b in Supplementary Fig. 9 exceeds the data point as mentioned by the green line. Please clarify it more clearly.

Q8. In Fig. 5a, the authors provided theoretical efficiency limitations DSC for different intensities of the model Osram 930 Warm White fluorescent tube light. It is interesting to know whether or not the authors considered the voltage loss in case of DSC and have calculated theoretical efficiency limitations from SQ-limit.

Reviewer #3 (Remarks to the Author):

The authors report an organic photosensitizer, coded MS5, that together with another dye (XY1b) forms a pancreatic system enabling an average power conversion efficiency (PCE) of 13.5% using a Cu(II/I)-based electrolyte (and a maximum Voc of 1.24 V). The study is very well performed and contain results from a series of dyes as well as a co-sensitization using two of the dyes. Optical, electronic, and electrochemical properties are reported together with synthesis details and ^1H and ^{13}C NMR spectra of the new compounds (in the SI). Transient photovoltage decay and charge extraction, nano-second flash photolysis, and electrochemical impedance spectroscopy are used together with the optical data and related to the quasi Shockley-Queisser limit and transport limit. The study also include characterization of the solar cells under indoor conditions, with an impressive PCE of 34.5% and 0.98 V. All of the above give noteworthy results that have high significance to the field. All in all, a very well performed and written contribution that I recommend to be published with only some minor reformulations as indicated below.

Some formulations are in my opinion too strong and give the wrong impression, with for example "...corresponds to an unprecedented PCE of 34.5%, outperforming the previous record set by a DSC28 with a smaller area of 0.25 cm^2 ..." in at line 276. The study (ref 28) reports 34.0% at 1000 lux for a small area device using also a panchromatic co-sensitization, a Cu (II/I)-based electrolyte ,and also report larger area cells ($3.2 - 8 \text{ cm}^2$), some of the larger area cells with 33% efficiency. The uncertified PCE values and small differences compared to previous studies cannot be taken to be exact, as also shown for the 500 lux illumination where e.g. ref 28 show 32.7% PCE compared to the lower 32.3% reported in the present study. The formulation that the study "outperform the previous record" is thus not true and the formulations also give the false impression that high indoor efficiencies for DSCs has only been reported for small area cells before." I urge the authors to reformulate this to more modestly compare with previous studies as these values are very close to each other (especially as they are recorded in uncertified laboratories), and also let the reader know that DSCs with higher areas that the ones reported in this study have PCEs in the 30-33% range.

Reply to REVIEWER COMMENTS

Reviewer #1 (Remarks to the Author):

This paper describes the synthesis, optical, electrochemical, and photovoltaic properties of novel push-pull dyes, MS4 and MS5 for copper(I/II) redox couple-based dye-sensitized solar cells (DSCs). The authors achieved a highest open-circuit voltage (V_{oc}) of 1.24 V using MS4 dye. Moreover, a combination of MS5 and XY1b dyes led to a remarkable power conversion efficiency (PCE) of 13.5% under AM1.5 condition and a PCE of 34.5% under ambient light condition. Such photovoltaic performances are very impressive and may attract considerable interest from researchers in the field of dye-sensitized solar cells (DSSCs). However, I have to raise a couple of concerns that should be adequately addressed before the publication.

1) I cannot find the IR and melting point of the new compounds in the supporting information. The data of NMR, high-resolution mass spectrum, IR, and melting point are necessary to identify the structures of the new compounds.

Answer: We added the data in the Supplementary Information.

2) The absorption peaks of MS4 and MS5 on TiO_2 appear at 470 nm in Fig. 1, whereas the corresponding peaks of IPCE values are seen at 520 nm. The discrepancy should be rationalized.

Answer: Note that the absorption spectra are different from IPCE spectra. The absorption spectra of samples in Fig. 1 were obtained by using dye-sensitized TiO_2 films with thickness of 2.2 μm in contact with atmosphere during measurements. In fact, as shown in the figure below, when we compared the difference of absorption spectra between MS5/ TiO_2 film with thickness of 2.2 μm and MS5/ TiO_2 with thickness of 8.0 μm in conjunction with the Cu(II/I) electrolyte, the absorption peaks of both samples are located at the same wavelength of 468 nm. The IPCE spectra in Fig. 2 were measured with full devices where the dye-sensitized 8.0 μm thick TiO_2

films are infiltrated by the Cu(II/I) electrolyte which absorbs some visible light. Moreover, the counter-electrode reflects light of longer wavelengths that is transmitted by the dye sensitized film back into the device enhancing the IPCE in the red spectral region. Finally, the IPCE values depend on the efficiency of light harvesting charge generation, and charge collection. This explains the difference between the IPCE and the absorption spectra.

UV-vis absorption spectra of MS5 adsorbed on 2.2 μm thick TiO₂ films (black line), and MS5 adsorbed on 8.0 μm thick TiO₂ films in conjunction with Cu (II/I) electrolyte (red line).

3) The oxidation potentials were determined by cyclic voltammetry (Supplementary Fig. 2). The values listed in lines 107-110 do not seem to match those taken from the Fig. 2. Moreover, the electrochemical processes are irreversible or quasi-irreversible. They should use differential pulse voltammetry to determine the oxidation potentials more accurately.

Answer: We thank the reviewer for proposing differential pulse voltammetry (DPV) to determine the oxidation potential of the new sensitizers. We have followed the suggestion and have performed the additional DPV measurements, results being reported in Supplementary Fig. 2 i-l. These confirm the E° data obtained from the CV experiments.

We note nevertheless that cyclic voltammetry appears to be the most suitable electrochemical method for determining the standard electrode potential (E°) of our dye molecules attached to the surface of mesoporous TiO_2 films. The disadvantage of differential pulse voltammetry (DPV) is that so far no theory has been developed for surface immobilized electroactive species allowing to determine the E° accurately by this technique. Thus, even though DPV measurements on surface adsorbed molecules have been reported previously, showing the maximum of the DPV current to be located close to E° their interpretation is not straightforward (*Reference R1*).

On the other hand, DPV is very useful to determine the E° of dissolved species under Nernstian (electrochemically reversible) kinetic conditions. A suitable criterion for Nernstian behaviour of solution species is that the peak width at half height $W_{1/2}$ is equal to: $W_{1/2}=3.52RT/nF$, which amounts to $W_{1/2}= 90.54/n$ mV at $T = 298$ K. The formula relating the peak maximum potential (E_{max}) and E° is $E_{\text{max}} = E^\circ + (RT/nF) \ln\{(D_{\text{Red}}/D_{\text{Ox}})^{1/2}\} - \Delta E/2$, where ΔE is the pulse height, D_{Red} and D_{Ox} are the diffusion coefficients of Red and Ox, R is the universal gas constant, T is the absolute temperature, n is the number of electrons for the reaction $\text{Ox} + ne \rightleftharpoons \text{Red}$, and F is the Faraday constant (*Reference R2*). If D_{Red} and D_{Ox} have similar values, $E_{\text{max}} \cong E^\circ - \Delta E/2$.

On the other hand, cyclic voltammetry (CV) is a well-established technique for determining E° for both solution dissolved and surface-attached species. In the case of Nernstian reactions with surface-attached species, the peak potential for both the anodic ($E_{\text{max(an)}}$) and the cathodic ($E_{\text{max(cat)}}$) branch of the voltammogram is equal to E° : $E_{\text{max(an)}} = E_{\text{max(cat)}} = E^\circ$. The equation applies either to monolayers as well as thicker layers if surface diffusion is fast on the time scale of the experiment. For quasi-reversible systems, with slow electrode kinetics, as it is the case for our metal oxide-attached dyes one finds $E_{\text{max(an)}} > E^\circ > E_{\text{max(cat)}}$. However, even a Nernstian system can appear as quasi-reversible, with $E_{\text{max(an)}} - E_{\text{max(cat)}} > 0$, in presence of a high Ohmic resistances causing a substantial voltage drop between the working and reference electrode due to high electrolyte and electrode resistance. The latter is the case for our dye sensitized mesoscopic TiO_2 films that are supported on a conducting FTO glass as current collector exhibiting a high sheet resistance.

In the initial version of the manuscript, we obtained the HOMO energies from the onset potential of the anodic CV peak, the method being widely used in polymers solar cell applications (*Reference R3*), other than the average of anodic and cathodic peak potential. Therefore, the values listed in the main text do not seem to match those taken from the Supplementary Fig. 2. We have amended this in the revised manuscript where we introduce the reasonable approximation that the deviation from Nernstian behaviour is mostly due to ohmic losses, hence

$$E^{\circ} = E_{\max(\text{cat})} + [E_{\max(\text{an})} - E_{\max(\text{cat})}] \frac{I_{\max(\text{an})}}{I_{\max(\text{an})} + |I_{\max(\text{cat})}|}$$

(*Reference R4*) where $I_{\max(\text{an})}$ and $I_{\max(\text{cat})}$ are the anodic and cathodic peak currents. The E° value determined above is the intercept of the line connecting the two peaks with the axis of zero current. E° values are tabulated vs. ferrocene and vs. the standard hydrogen electrode (SHE) in Supplementary Table 1. The latter are determined by adding the conversion factor 0.624 V based on the proposition by Pavlishchuk and Addison that in acetonitrile (*Reference R5*).

As stated above, in the revised version, we recorded DPV curves and remeasured the CV curves. The CV and DPV curves are displayed separately for better visualisation, and discussed in the legend to Supplementary Fig. 2.

References:

- R1. Brown A. P. & Anson F. C. Cyclic and Differential Pulse Voltammetric Behavior of Reactants Confined to the Electrode Surface. *Anal. Chem.* **49**, 1589–1595 (1977).
- R2. Bard A. J. & Faulkner L. R. *Electrochemical Methods*. 2nd Edition, Wiley, 2001.
- R3. Cardona, C. M., Li, W., Kaifer, A. E., Stockdale, D. & Bazan, G. C. Electrochemical Considerations for Determining Absolute Frontier Orbital Energy Levels of Conjugated Polymers for Solar Cell Applications. *Adv. Mater.* **23**, 2367–2371 (2011).
- R4. Vallot R., N'diaye A., Bermont A., Jakubowicz C., & Yu L.T., Comportement electrochimique en milieu aqueux de composes organiques moleculaires insolubles ou peu solubles—I. Voltamperometrie de derives quinoniques. *Electrochim. Acta* **25**, 1501–1512 (1980).

R5. Pavlishchuk V. V. & Addison A. W. Conversion constants for redox potentials measured versus different reference electrodes in acetonitrile solutions at 25 °C. *Inorganica Chimica Acta* **298**, 97–102 (2000).

Reviewer #2 (Remarks to the Author):

In this study, two high-Voc organic dyes of MS4 and MS5 have been designed and synthesized. The highest Voc of 1.24V was obtained by MS5 and co-sensitization with XY1b achieved an efficiency of 13.5% under standard AM1.5G sunlight condition. Also, the DSC with the active area of 2.8 cm² achieved a record PCE of 34.5% under standard Osram 930 Warm White fluorescent tube light at 1,000 lux intensity. This is an extremely good work in DSSCs. Thus, I recommend it for publication in Nature Communications after a minor revision. Some issues need to be addressed:

Q1. In Supplementary Fig. 1, MS4 and MS5 dyes displayed their highest absorption at around 330 nm. Thus, in Supplementary Table 1, these values around their λ_{\max} (330nm) should be added.

Answer: We added the data in the Supplementary Table 1 with red color highlighting.

Q2. In the third paragraph of 'Results and discussion', the authors addressed that the higher IPCE of MS4 and MS5 indicates a more efficient charge collection yield. However, as far as we can see, the value of IPCE can be also influenced by the light harvest efficiency and charge injection efficiency, as well as the charge regeneration efficiency. Please clarify it more clearly.

Answer: The IPCE is determined by the light harvest efficiency, the charge injection efficiency, the charge regeneration efficiency, and the charge collection efficiency. In the manuscript, we discussed the comparison of IPCE spectra among MS4, MS5 and NT35. Compared to MS4 and MS5, NT35 has a higher absorption spectra and favorable molecular energy level offset at the interfaces of dye/TiO₂ and dye/electrolyte, which can lead to a higher light harvest efficiency, the charge injection efficiency, and the charge regeneration efficiency. The lower IPCE peak value of NT35 most probably stemmed from a lower charge collection yield. This was further confirmed by transient photovoltage decay measurement showing that the NT35 device has

shorter electron lifetimes than MS4 and MS5. For clarity, we modified the discussion in the manuscript with red colour highlighting.

Q3. “MS5+XY1b”-based DSC exhibits excellent photostability and maintained 93% of its initial value during light soaking at full solar intensity for 1,000 hours. How could the device be stable up to 1000 hours with only a little PCE loss? Please provide how to encapsulate the device in more details.

Answer: The devices were first sealed with ThreeBond UV light curing glue and then further sealed with 3M™ Scotch Weld™ Epoxy Adhesive DP460, which can prevent the leakage of electrolyte and also prevent the atmosphere gas (water, O₂) from penetrating into the devices. We detailed the encapsulation method with red colour highlighting in the experimental section (Characterization of Solar Cells) in the Supplementary Information.

Q4. The dye-loading amounting is $2.70 \times 10^{-8} \text{ mol cm}^{-2} \mu\text{m}^{-1}$ for “MS5+XY1b”. Is it possible to also show the dye-loading amount of each dye of MS5 and XY1b, respectively?

Answer: We added the values of dye loading amount of each dye in co-sensitization system in the revised manuscript with red colour highlighting.

Q5. We noticed that the authors used different concentrations of copper electrolyte for application in AM1.5G sunlight and ambient light. Thus, how will concentration change affect the efficiency of DSSCs, especially in ambient light?

Answer: The DSCs performing at ambient light conditions can employ electrolytes with lower concentrations of Cu(I) and Cu(II), since the photocurrents are smaller. We added the discussion of the effect of electrolytes on the efficiency of DSSCs in ambient light in the modified manuscript with red colour highlighting. Additional data were tabulated in Supplementary Table 4.

Q6. According to the authors in line number 63, the authors addressed “Benefiting from the high Voc of one dye and the high Jsc of the other, co-sensitized solar cell can reach a higher PCE and stability than the one employing either one of dyes”. It seems that the band gap of dye can

determine the V_{oc} . Generally speaking, co-adsorption is effective when two dyes have supplementary absorption spectra, leading to a high J_{sc} . Please clarify them more clearly.

Answer: The co-sensitization was introduced in DSC to improve J_{sc} when the dyes have supplementary absorption spectra. In recent development of highly efficient DSSCs, the organic dyes can achieve a high molar extinction coefficient and panchromatic absorption. However, the V_{oc} is limited. Base on the co-sensitization of the panchromatic dye with a narrow absorptive, high V_{oc} dye, the device has a higher V_{oc} and PCE than the one with the panchromatic dye alone. We clarified this point in the modified manuscript with red colour highlighting.

Q7. According to the equation 1 and n value of “MS5+XY1b” in Supplementary Fig. 9, it was calculated that the V_{oc} of “MS5+XY1b” should not be higher than 0.88 V at 1000 lux ($318.2 \mu\text{W cm}^{-2}$), as V_{oc} shows a linear relationship with $\ln I$. But, in this report, the authors found the V_{oc} of MS5+XY1b around 0.98 V. Furthermore, the lowest value of light intensity in case of MS5+XY1b in Supplementary Fig. 9 exceeds the data point as mentioned by the green line. Please clarify it more clearly.

Answer: Note that the devices measured in 1000 lux have different electrolyte composition from the ones measured under AM1.5G full sunlight conditions. In Supplementary Fig. 9, the devices measured under AM1.5G sunlight conditions used an electrolyte containing 0.2 M $[\text{Cu(I)(tmby)}_2]\text{TFSI}$ and 0.1 M $[\text{Cu(II)(tmby)}_2](\text{TFSI})_2$ with 0.1 M LiTFSI and 0.6 M NMB in acetonitrile. However, the devices measured in 1000 lux employed 0.1 M $[\text{Cu(I)(tmby)}_2]\text{TFSI}$ and 0.04 M $[\text{Cu(II)(tmby)}_2](\text{TFSI})_2$ cs with 0.1 M LiTFSI and 0.6 M NMB in acetonitrile. The plot of ideality factor of the indoor device is shown in Supplementary Fig. 12. The ideality factors of these two kinds of devices are different. We clarified this point in the manuscript with red colour highlighting. Also, we added the discussion of the effect of electrolyte concentrations on device performance in the manuscript with red colour highlighting.

Q8. In Fig. 5a, the authors provided theoretical efficiency limitations DSC for different intensities of the model Osram 930 Warm White fluorescent tube light. It is interesting to know whether or not the authors considered the voltage loss in case of DSC and have calculated theoretical efficiency limitations from SQ-limit.

Answer: The calculation shown in Fig. 5a is a Shockley-Queisser (SQ) limit calculation with the Osram 930 spectrum instead of the AM1.5 that is usually used. It is assumed that:

- Complete absorption above bandgap, zero absorption below bandgap and zero charge transport losses (to calculate J_{sc})
- Only radiative recombination (to calculate V_{oc})
- Ideal single diode behavior (i.e. ideality factor of 1) to calculate the $I-V$ curve from which the efficiency is determined.

The SQ limit does not account for additional voltage losses that might occur in a specific solar cell realization. For clarity, we modified the discussion in the manuscript with red colour highlighting.

Reviewer #3 (Remarks to the Author):

The authors report an organic photosensitizer, coded MS5, that together with another dye (XY1b) forms a pancreatic system enabling an average power conversion efficiency (PCE) of 13.5% using a Cu(II/I)-based electrolyte (and a maximum V_{oc} of 1.24 V). The study is very well performed and contain results from a series of dyes as well as a co-sensitization using two of the dyes. Optical, electronic, and electrochemical properties are reported together with synthesis details and 1H and ^{13}C NMR spectra of the new compounds (in the SI). Transient photovoltage decay and charge extraction, nano-second flash photolysis, and electrochemical impedance spectroscopy are used together with the optical data and related to the quasi Shockley-Queisser limit and transport limit. The study also include characterization of the solar cells under indoor conditions, with an impressive PCE of 34.5% and 0.98 V. All of the above give noteworthy results that have high significance to the field. All in all, a very well performed and written contribution that I recommend to be published with only some minor reformulations as indicated below.

Some formulations are in my opinion too strong and give the wrong impression, with for example "...corresponds to an unprecedented PCE of 34.5%, outperforming the previous record set by a DSC28 with a smaller area of 0.25 cm^2 ..." in at line 276. The study (ref 28) reports 34.0% at 1000 lux for a small area device using also a panchromatic co-sensitization, a Cu (II/I)-based electrolyte, and also report larger area cells ($3.2\text{--}8 \text{ cm}^2$), some of the larger area cells with 33% efficiency. The uncertified PCE values and small differences compared to previous studies cannot be taken to be exact, as also shown for the 500 lux illumination where e.g. ref 28 show 32.7% PCE compared to the lower 32.3% reported in the present study. The formulation that the study "outperform the previous record" is thus not true and the formulations also give the false impression that high indoor efficiencies for DSCs has only been reported for small area cells before." I urge the authors to reformulate this to more modestly compare with previous studies as these values are very close to each other (especially as they are recorded in uncertified laboratories), and also let the reader know that DSCs with higher areas that the ones reported in this study have PCEs in the 30-33% range.

Answer: We thank the reviewer for the helpful comments. We modified the discussions according to his suggestions in the manuscript as shown with red colour highlighting.

REVIEWERS' COMMENTS

Reviewer #1 (Remarks to the Author):

The revised manuscript addresses the reviewer's concerns and is improved significantly compared to the original version. I recommend the publication.

Reviewer #2 (Remarks to the Author):

The authors have already revised the manuscript according to the comments of the Referees. It can be acceptable in Nature Communications without further revision.

Reviewer #3 (Remarks to the Author):

The authors have successfully revised their manuscript with respect to mine and the other reviewers' comments and concerns. The manuscript is now ready for publication in my opinion.